# Identification of R-Spondin Gene Signature Predictive of Metastatic Progression in BRAFV^600E^-Positive Papillary Thyroid Cancer

**DOI:** 10.3390/cells12010139

**Published:** 2022-12-29

**Authors:** Sabrina Daniela da Silva, Grégoire B. Morand, Luciana Diesel, Jefferson Muniz de Lima, Krikor Bijian, Senthilkumar Kailasam, Francois Lefebvre, Guillaume Bourque, Michael Hier, Moulay A. Alaoui-Jamali

**Affiliations:** 1Department of Otolaryngology—Head and Neck Surgery, Sir Mortimer B. Davis-Jewish General Hospital, McGill University, Montreal, QC H3T 1E2, Canada; 2Lady Davis Institute for Medical Research/Segal Cancer Centre, Departments of Medicine and Oncology, Sir Mortimer B. Davis-Jewish General Hospital, McGill University, Montreal, QC H3T 1E2, Canada; 3Department of Otorhinolaryngology—Head and Neck Surgery, Luzerner Kantonsspital, 6004 Lucerne, Switzerland; 4Department of Otorhinolaryngology-Head and Neck Surgery, University Hospital Zurich, Frauenklinikstrasse 24, 8091 Zurich, Switzerland; 5Canadian Centre for Computational Genomics, McGill University, Montreal, QC H3A 0G1, Canada; 6Department of Human Genetics, McGill University, Montreal, QC H3A 0G1, Canada

**Keywords:** thyroid cancer, papillary, R-Spondin 4 (RSPO4), BRAF, prognosis

## Abstract

Papillary thyroid carcinoma (PTC) is the most common malignancy of the thyroid gland and early stages are curable. However, a subset of PTCs shows an unusually aggressive phenotype with extensive lymph node metastasis and higher incidence of locoregional recurrence. In this study, we investigated a large cohort of PTC cases with an unusual aggressive phenotype using a high-throughput RNA sequencing (RNA-Seq) to identify differentially regulated genes associated with metastatic PTC. All metastatic PTC with mutated BRAF (V600E) but not BRAF wild-type expressed an up-regulation of R-Spondin Protein 4 (RSPO4) concomitant with an upregulation of genes involved in focal adhesion and cell-extracellular matrix signaling. Further immunohistochemistry validation confirmed the upregulation of these target genes in metastatic PTC cases. Preclinical studies using established PTC cell lines support that RSPO4 overexpression is associated with BRAF V600E mutation and is a critical upstream event that promote activation of kinases of focal adhesion signaling known to drive cancer cell locomotion and invasion. This finding opens up the potential of co-targeting B-Raf, RSPO and focal adhesion proteins as a pharmacological approach for aggressive BRAF V600E PTC.

## 1. Introduction

Thyroid cancer is the most common endocrine malignancy worldwide [1]. Among the four differentiated histological types, papillary thyroid cancer (PTC) represents 85% of thyroid gland cancers [2]. Epidemiological studies have shown gradual increased in PTC incidence over the past decades, particularly among young patients and women [3,4]. The main risk factors include exposure to radiation [5] and inherent genetic factors related to family history and ethnicity [4]. Although early stages of PTC are curable and the 10-year overall survival rates are >97%, there is a subgroup of high-risk PTC patients presenting frequently with lymph node metastasis [6], distant metastasis with high tumor recurrence rate [7,8,9].

PTC are staged according to the American Joint Committee on Cancer (AJCC) TNM system. Few years ago, the American Thyroid Association (ATA) proposed a new three-tiered clinicopathological risk stratification system that classified patients as having a low, intermediate, or high risk of recurrence [10]. However, both systems do not adequately predict individual risk of recurrence or metastasis [11]. Methods to predict and/or stratify the risk of PTC recurrence are crucial to avoid overtreatment of low-risk and undertreatment of high-risk patients. It will also help to identify those patients who are at risk of recurrence over time in order to tailor their treatment plan accordingly.

Multi-omics experiments in thyroid cancer have allowed a comprehensive analysis and understanding of the molecular changes linked to tumor progression [12,13,14,15,16]. Mutations in BRAF and in particular V600E are found in approximately 50% tumors, but there is no consensus on whether this biomarker is an independent predictor for recurrence risk of PTC [17,18,19,20]. Consequently, the 2015 ATA guideline does not routinely recommend BRAF V600E alone for initial postoperative risk stratification [11]. However, an increasing number of studies show that genetic signatures may supersede tumor size and other classical staging criteria in predicting the presence of aggressive pathologic features in PTC [21,22].

PTC recurrence is estimated to occur at a rate of 4% to 28% [9]. The primary goal of this study was to identify transcriptomic differences in a unique cohort of rare PTCs with an unusual aggressive phenotype presenting with extensive lymph node metastasis and higher incidence of locoregional recurrence (metastatic), compared to non-metastatic PTC. All the metastatic PTC had BRAF mutation and up-regulation of R-Spondin protein (RSPO4). Genes specifically deregulated in BRAF-mutated PTC was investigated to determine their implication in the metastatic signaling and further validated in a large independent set of PTC samples. Preclinical studies using a panel of metastatic PTC cell lines supported that RSPO4 overexpression associated with BRAF V600E mutation can promote activation of focal adhesion kinases signaling in metastatic PTC.

## 2. Materials and Methods

### 2.1. Study Population

After Ethics Review Board approval (#13-093), patients diagnosed with PTC were prospectively enrolled in the fresh tissue biobank of Sir Mortimer B. Davis-Jewish General Hospital in Montreal, QC, Canada. Patients were consented at the time of surgery for use of their samples. Full access to their medical records was granted for research purposes. Detailed demographics (clinical stage, lymph nodes involvement, histological subtype and variant, extra-thyroidal extension, multifocality, and surgical margins), treatment, follow-up and recurrence data were obtained retrospectively. Clinicopathological data were handled in a coded fashion according to ethical guidelines. Exclusion criteria were papillary micro-carcinoma, pregnancy, age less than 18 years, and missing follow-up. Strengthening the reporting of observational studies (STROBE Statement) was used to ensure appropriate methodological quality (http://www.strobe-statement.org/, accessed on 1 January 2021).

### 2.2. Sample Preparation and RNA Isolation

RNA samples were obtained from a unique cohort of 20 fresh-frozen PTCs (10 metastatic vs. 10 non-metastatic) followed-up for over than 10 years. Fresh-frozen tumor material was collected at the time of surgical resection, snap-frozen in liquid nitrogen, and stored at −80 °C until RNA extraction. Time between tumor resection and storage in liquid nitrogen did not exceed 45 min. Metastatic and non-metastatic samples were matched for age, sex and tumor stage (TNM) to assure comparability of the two groups (Table 1). RNA extraction was done using mRNeasy Kit (Qiagen, Inc., Valencia, CA, USA) according to the manufacturer’s instruction. All RNA samples were assessed for quality using the RNA 6000 Nano assay on the 2100 Bioanalyzer (Agilent Technologies, Inc., Santa Clara, CA, USA) and for quantity by Nanodrop (Thermo Scientific, Wilmington, DE, USA).

### 2.3. RNA Library Construction and Sequencing

RNA library construction, sequencing and bioinformatics analysis were performed by the McGill University and Genome Quebec Innovation Centre. rRNA depletion was performed on 200–400 ng of each total RNA sample with the RiboZeroGold (Illumina, San Diego, CA, USA) as per the manufacturer’s instructions. The entire rRNA depleted fraction (ranging 4–22 ng) was used as input for library preparation using the ScriptSeq V2 library preparation kit (Illumina, San Diego, CA, USA). All libraries were validated and quantified with the Bioanalyzer DNA 1000 assay (Agilent Technologies, Inc., CA, USA) and further quantified with the Qubit DNA Broad Range assay (Life Technologies, Carlsbad, CA, USA). 10 μL of each library were diluted to a concentration of 10 nM. Equal volumes of each 10 nM library were then pooled for subsequent paired-end sequencing on an Illumina HiSeq 2000/2500 (Illumina, San Diego, CA, USA). Sequencing was performed with 4 samples per lane, hence generating 62 to 106 million paired end reads per library. Base calls were made using the Illumina CASAVA pipeline. Base quality was encoded in phred 33.

### 2.4. Bioinformatic Analysis

Sequence alignment and quantification of exon expression were carried out using an internally developed RNA-Seq analytical pipeline [23] Reads were trimmed from the 3′ end to have a phred score of at least 30. Illumina sequencing adapters were removed from the reads, and all reads were required to have a length of at least 32. Trimming and clipping were done with the Trimmomatic software [24]. The filtered reads were aligned to a reference genome (hg19). The alignment was done with the combination of tophat/bowtie software [25]. The Cufflinks program [26] was used to assemble aligned RNA-Seq reads into transcripts and to estimate their abundance (FPKM). Several metrics and exploratory analysis to control the data quality and to verify the biological reliability of the data were used and are presented in the first section of the results. The differential gene expression analysis was done using DESeq [27] and edgeR [28] R Bioconductor package. Gene ontology (GO) and Kyoto Encyclopedia of Genes and Genomes (KEGG) enrichment analyses was implemented using the clusterProfiler package [29].

### 2.5. Cell Culture

The papillary thyroid cancer cell line TPC-1 (ATCC, Manassas, VA, USA) and the metastatic anaplastic thyroid cancer cell lines (8505c and THJ-16T) were maintained in RPMI medium (Corning, Manassas, VA, USA ) supplemented with 10% heat-inactivated fetal bovine serum (FBS) (Mediatech Inc., Herndon, VA, USA) and 50 U/mL of penicillin-streptomycin (1%). Cells were cultured at 37 °C with 5% CO_2_. All cell lines were authenticated using STR profiling. Cell lines were routinely treated with MycoZAP (Lonza, NJ, USA) and tested for mycoplasma contamination.

### 2.6. CRISPR-Cas9 for the Generation of RSPO4 and BRAF Knockout Cells

Five target guide sequences for human RSPO4 and BRAF knockdown (Table 2) were cloned into lentiCRISPRv2 vector. Metastatic and non-metastatic thyroid cancer cell lines (8505c, THJ-16T were then transduced by lentiviral particles. The wide-type control was only the lentiCRISPR vector. The transduced cells were selected with puromycin at 1 µg/mL. RSPO4 and BRAF knockout clones were confirmed by qRT-PCR and immunoblotting.

### 2.7. qRT-PCR

Total RNA was isolated from control and knockdown THJ-16T, 8505c, and TPC-1 cell lines. cDNA was synthesized from 500 ng of RNA using Superscript II reverse transcriptase (Invitrogen^®^, Carlsbad, CA, USA) and oligo-dT primers (Invitrogen^®^, CA, USA ). qRT-PCR was be performed in the ABI PrismTM 7900 (Applied Biosystems^®^, Foster City, CA, USA) using SYBR^®^ Green (Applied Biosystems^®^, Foster City, CA, USA) in a 10 μL total volume and quality controls were used as proposed by MIQE Guidelines [30]. Specific primer set sequences was designed (Table 3). The reactions were carried out in triplicate. GeNorm (https://genorm.cmgg.be/, accessed on 1 January 2022) algorithm were used to determine the most stable genes. The software packages were used as excel add-ons. *ACTB*, *GAPDH* and *HRPT1* was selected by the algorithm as endogenous control. Fold differences in the relative gene expression were calculated using Pfaffl model [31].

### 2.8. Western Blot Analysis

Total cell extracts from exponentially growing cells were collected by scrapping into modified radioimmunoprecipitation assay lyses buffer (RIPA) supplemented with 20 mg/mL pepstatin A, 1 mM PMSF, and protease inhibitor cocktail (Roche, Indianapolis, IN, USA). Blots were detected using specific antibodies for: anti-Rspo4 (1:20,000; Abcam, Cambridge, MA, USA), anti-Braf (1:1000; Santa Cruz Biotechnology, Dallas, TX, USA), anti-N-cadherin (1:1000; BD Biosciences, San Jose, CA, USA), anti-β-catenin (1:1000; Cell Signaling Technology, Danvers, MA, USA), anti-Vimentin (1:500; BD Biosciences, San Jose, CA, USA), anti-c-Myc (Y69; 1:1000; Abcam, Cambridge, MA, USA), anti-Twist (1:1000; Abcam, Cambridge, MA, USA), anti-Snail (1:500; Cell Signaling Technology, Danvers, MA, USA), anti-FAK (1:500; Upstate Biotechnology, Inc. New York, NY, USA), anti-p-Y861-FAK (1:1000; GenScript Corp., Hong Kong, China), anti-Src (1:2000; Upstate Biotechnology, Inc. New York, NY, USA), anti-p-Src (1:500; Cell Signaling Technology, Danvers, MA, USA), anti-Paxilin (1:1000; MilliporeSigma, Rocklin, CA, USA), anti-p-Y118-Paxilin (1:1000; Invitrogen®, CA, USA), and anti-Gapdh (internal control; 1:10,000; MilliporeSigma, Rocklin, CA, USA) at 4 °C overnight. The signal was detected with horseradish peroxidase (HRP)-conjugated secondary antibodies diluted at 1:5000 for 1 h and developed with an enhanced chemiluminescence detection system (PierceTM ECL Western Blotting Substrate, Thermo Fisher Scientific Inc, Cleveland, OH, USA) according to manufacturer’s instructions.

### 2.9. Cell Proliferation

Cell proliferation was evaluated in THJ-16T, 8505c, TPC-1 (RSPO4 knockout and control cells) using the MTT 3-(4,5-dimethylthiazol-2-yl)-2,5-diphenyltetrazolium bromide metabolic assay as we previously described [29]. Cell morphology was evaluated under inverted microscope at 100× of magnification. Briefly, 3 × 10^3^ cells were then seeded in 96-well-plates in 12 replicates. 12 h, 24 h, and 48 h of incubation, 0.5 mg/mL MTT was added to each well. After 3 h incubation at 37 °C, formazan crystals were dissolved with 4 mM HCl, 0.1% Nonidet P-40 in isopropanol and the supernatant absorbance was determined at 570 nm using a microplate reader Fluostar Optima (BMG LabTech, Jena, Germany). Results are expressed as mean ± SD of at least 3 independent experiments. Statistical analysis was done using Student’s *t* test.

### 2.10. Cell Motility and Invasion Assays

Invasion was measured by evaluating the migratory cell rate through a polycarbonate membrane (8-μm pore diameter) coated with BD Matrigel Matrix (BD Biosciences, Bedford, MA, USA) in a modified Boyden chamber (Corning, cat. no. 3422) as previously described [30]. Collective cell migration was evaluated using qualitative wound-healing assay [30]. Three independent experiments were performed for each condition and results were expressed as mean ± SD. Statistical analysis was done using Student’s *t* test.

### 2.11. Immunohistochemistry (IHC) in the PTC Tissue Microarray (TMA)

Immunohistochemistry reaction was carried out on the TMA with 171 samples [32]. In brief, the slides were incubated with primary antibodies diluted in PBS overnight at 4 °C using: anti-RSPO4 (PA525615, Thermo Fisher Scientific Inc, Cleveland, OH, USA; 1:50) and anti-BRAF (20899-1-AP, Proteintech Group Inc, Chicago, IL, USA; 1:200). Sections were incubated with secondary antibodies (Advanced TM HRP Link, DakoCytomation, Glostrup, Denmark) for half-hour followed by the polymer detection system (Advanced TM HRP Link, DakoCytomation, Glostrup, Denmark) for half-hour at room temperature. Reactions were developed using a solution of 0.6 mg/mL of DAB (Sigma, St Louis, MO, USA) and 0.01% H_2_O_2_ and then counter-stained with hematoxylin. Positive controls were included in all reactions in accordance with manufacturer’s recommendations. Negative control consisted in omitting the primary antibody and replacing the primary antibody by normal serum. IHC reactions were replicated on distinct TMA slides to represent different tissues levels in the same lesion. The second slide was 25–30 sections deeper than the first slide, resulting in a minimum of 300 μm distance between sections representing 4-fold redundancy with different cell populations for each tissue.

Two independent certified pathologists conducted the IHC analysis blindly to the clinical data. Cores were scanned in 10× power field to settle on the foremost to marked area predominant in a minimum of 10% of the neoplasia [32,33]. IHC reaction was considered as positive if of a clearly visible dark brown precipitation occurred. IHC analysis considered the percentage and intensity of staining as: 0 (no detectable reaction or little staining in <10% of cells), 1 (weak but positive IHC expression in >10% of cells) and 2 (strong positivity in >10% of cells) [33].

### 2.12. Statistical Analysis

For in vitro analysis, statistical analyses were performed using the two-tailed Student’s *t*-test for unpaired samples to one independent experiment and one-way analysis of variance (ANOVA) with post hoc comparisons based on the Tukey’s multiple comparisons to two independent experiments. The analysis was carried on GraphPad Prism 5 software (GraphPad Software Inc., San Diego, CA, USA). Data were presented as mean ± SD, and the level of significance considered *p* ≤ 0.05. SD indicates dispersion of the data from mean and it was used to summarized the variability. For dichotomous variables, the two-sided Fischer Exact test was used to compare proportions. Odds ratio (OR) and 95% confidence interval (95% confidence interval; CI) were calculated according to the Mantel-Haenzel method. For discrete variables showing normal distribution, means and standard errors of means (SEM) are given and comparisons were made using the *t*-test. SEM was used to estimate the dispersion of the mean (average) of the data to be true among the population mean and it was limited to compute (CI) which measures the precision of population estimate. Alternatively, median, first quartile (Q25), and third quartile (Q75) are indicated and the non-parametric Kruskal–Wallis test was used. Statistical analyses were performed using SPSS^®^ 21.0.0 software (IBM©, Armonk, NY, USA).

## 3. Results

### 3.1. Patient Description

Fresh samples from patients with metastatic and non-metastatic PTC were matched for age, sex and T clinical stage (Table 1). The validation cohort constituted by a total of 171 patients with PTC, being 15 benign thyroid nodules and 20 PTC matched fresh samples (10 metastatic and 10 non-metastatic) and 136 PTCs with long-term follow-up. These patients had more than 10 years of follow-up. The mean age at diagnosis was 47 years (SD ± 14.6) with a total of 134 females (80.1%) and 37 males (19.9%). Total thyroidectomy was performed in 153 patients (89.5%), meanwhile hemithyroidectomy was performed in 18 cases (10.5%). Ninety-three (59.6%) of the patients had pT3–4 tumors, meanwhile 63 had pT1–2 (40.4%) tumors. Average tumor size was 2.6 cm (SD ± 1.53). Any extrathyroidal extension was noted in 76 (44.4%) cases. The N-stage was pN0 in 84 (53.8%) cases, pN1a in 35 (22.4%) cases, pN1b in 22 (14.2%) cases, and pNx in 15 (9.6%) cases, and not available in 15 cases (benign thyroid tissue cases) (Table 4).

### 3.2. Gene Expression Profiling of Metastatic vs. Non-Metastatic PTC

Differential expression analysis uncovered significant changes in gene expression between metastatic and non-metastatic PTC. Between 61 and 104 million 100-bp paired-end reads were generated for each sample with a mean mapping rate of 80%. The principal component analysis (PCA) demonstrated clear separation between the two groups (Figure 1A) considering the top DEGs (Figure 1B). Among 2081 protein coding genes identified to be differentially expressed between metastatic (ME) and non-metastatic (NM) cases, 824 were downregulated and 1257 were upregulated (<5% FDR, *p* < 0.05, log2FC > |2|) Figure 1B and Appendix A). Heatmap showing expression levels of the top genes that can differentiate metastatic from non-metastatic PTC samples (Figure 1C). Upregulated genes are shown in red and downregulated genes are shown in blue.

### 3.3. Functional Analysis of Genes Associated with Metastatic PTC

Function annotation of DEGs was done using clusterprofiler [29]. Gene set enrichment analysis (GSEA) was done using publicly available knowledge of databases such Gene Ontology (GO) terms [23] or Kyoto Encyclopedia of Genes and Genomes (KEGG) [24]. A *p*-value of ≤0.05 and FDR < 0.05 are considered strongly enriched. (Figure 2A). The GO and KEEG gene set enrichment pathways analysis revealed the largest gene ratio differences in molecular function (MF), cellular component (CC), biological process (BP) and signaling pathways are linked to cytokine-cytokine receptor interaction (CCRI) and focal adhesion signaling (FA) (Figure 2A**,**B). When we combined MF, CC, BP and KEEG analysis, FA genes were overlapped in more than one network, such as the cell adhesion molecules and ECM-receptor interaction pathways (Figure 2A). These genes were clearly able to distinguish between metastatic vs. non-metastatic groups (Figure 2B), suggesting that FA process had disrupted key genes in metastatic PTC. Hence, we decided to focus on this pathway and analyze the metastatic potential in PTC.

### 3.4. RSPO4 Is the Most Significantly Overexpressed Gene in Metastatic BRAF V600E PTC

In the unsupervisioned analysis (Figure 1C), a gene called *RSPO4* was highly overexpressed in metastatic PTC (<5% FDR, *p* < 0.05, log2FC > |2|). Based on the lists of pathways and networks potentially enriched in metastatic PTC (Figure 2A). *RSPO4* showed to be a promising candidate. *RSPO4* is a gene encoding a secretory protein that is a member of the R-spondin family of proteins [34]. Within this family, they share a common domain organization consisting of a signal peptide, cysteine-rich/furin-like domain, thrombospondin domain and a C-terminal basic region. There is no published study showing *RSPO4* up-regulation in aggressive thyroid cancer. The status and function of this protein is unknown in PTC. Though it has been found to cause anonychia congenital, a condition characterized by the absence of fingernails and toenails [34]. Public databases, literature and stringDB was explored to find genes associated with *RSPO4* in order to connect with the information from our results (Figure 3A). In addition to *RSPO4*, other family members (specifically *RSPO1* and *RSPO3*, not *RSPO2*) were differentially expressed in metastatic PTC, but only *RSPO4* showed the highest statistical significance that offered an unprecedented opportunity for further investigation (Figure 3B). RSPO proteins are known to bind to leucine-rich repeat-containing G-protein-coupled receptor (LGRs). In our study, it was confirmed that LGR4, LGR5, and LGR6 similarly had high expression (Figure 3B).

### 3.5. RSPO4 Can Serve as a Robust Biomarker of Metastatic PTC

To validate the differentially expressed genes that can discriminate between metastatic and non-metastatic PTC from our study, we used a public cancer database (The Cancer Genome Atlas—TCGA, Figure 4) to confirm our findings in an independent data set. The TCGA contains 496 PTC samples of which 240 are primary non-metastatic tumors (M0), 6 metastatic PTC (M1), and 165 cases were undefined regarding the metastatic status (MX). The survival analysis showed that high *BRAF* (Figure 4) expression was correlated with decreased survival rates both in metastatic and non-metastatic. (Figure 4). Though in the TCGA dataset *RSPO4* was also highly expressed in the metastatic samples (*p* < 0.005).

### 3.6. RSPO4 Overexpression Occurs Selectively in BRAF V600E PTC

Since both *RSPO4* and *BRAF* gene was highly expressed in our metastatic cohort (Figure 5A), we used Integrative Genome Viewer (IGV—https://software.broadinstitute.org/software/igv/, accessed on 1 January, 2022) to screened for common mutations described in thyroid cancer to identify those impacted on our target genes identified to discriminate between non-metastatic and metastatic PTC cases. Among our 20 samples, 10 harbored an A to T substitution at position 140,453,136 on chromosome 7, that is harboring a BRAF V600E classical mutation (Figure 5B). Interestingly, *BRAF* mutated samples showed overexpression of *RSPO4* (FPKM > 2 for *BRAF* V600E while FPKM < 1 for *BRAF* wt). Overall, the relative fold-change between *BRAF* V600E and *BRAF* wt was about 10 and statistically highly significant (*p* = 6.27 × 10^−7^). This result was confirmed by qPCR and IHC in 170 PTC samples (Figure 5C). In addition, the independent TCGA platform comparing *BRAF* V600E to *BRAF* wt papillary PTC found similar gene expression patterns (Figure 5). Our study shows that *BRAF* V600E mutation is associated with *RSPO4* at transcript and protein level.

### 3.7. BRAF Inactivation and RSPO4 Overexpression Influence PTC Cell Proliferation, Motility and Invasion

To evaluate the impact of *BRAF-RSPO4* in metastatic and non-metastatic conditions, both genes (*BRAF* and *RSPO4*) were deleted in different regions of the gene by using CRISPR technology (Figure 6A,B) as we previously described [35]. RSPO4 protein and mRNA expression was decreased after *BRAF* knockout when compared to wild-type (Figure 6A–D) in our panel of metastatic and non-metastatic human thyroid cancer cell lines (THJ-16T, 8505C and TPC-1). The CRISPR-RSPO4 and CRISPR-BRAF knockout carried out discrete changes in cellular morphology, however the knockout condition significantly impacted on the phenotype of metastatic and non-metastatic thyroid cancer cell lines by significantly decrease the proliferation, invasion, and migration status (Figure 6E–H).

### 3.8. RSPO4 Inhibition in PTC Cells Prevents the Activation of Proteins Involved in Focal Adhesion Signaling

Metastatic and non-metastatic thyroid cancer cell lines carrying CRISPR-RSPO4 knockout, and their controls were used to understand how the gene modulation could influence the phenotypic changes related to cell movement and invasion. As expected, inhibition of RSPO4 induced a downregulation of Wnt/catenins and an upregulation of E-cadherins. Interestingly, RSPO4 downregulation resulted in a potent inhibition of activated forms of mutiple phosphorylated proteins involved in focal adhesion signaling (Figure 7A–C). These include Y397 (p-PTK2), the autophosphorylation site of focal adhesion kinase (PTK2/FAK) critical for PTK2 activation and interaction with downstream partners such as 118Y-p-Paxillin, an adapter proteins and partner of PTK2 in focal adhesion signaling and regulation of focal adhesion turnover during cancer cell locomotion; and 416Y-p-Src, which regulate Src multiple downstream signaling pathways, including Src interaction with PTK2 during in cell migration and invasion.

## 4. Discussion

Genomic testing of thyroid cancer nodules is becoming an important component in decision-making in clinical care, reducing the need for diagnostic surgery and improving accuracy of cancer risk assessment. Despite advances in molecular testing and the discovery of new promising therapeutics, effective treatments for advanced radioactive iodine (RAI)-refractory metastatic differentiated thyroid cancer (DTC) are still lacking. This study focused on the identification of novel biomarkers for aggressive PTC using a cohort of PTC cases with over than 10 years fellow-up and characterized by an unusual invasiveness, including extensive lymph node metastases and higher incidence of locoregional recurrence.

2034 differentially expressed genes (DEGs) were identified by high-throughput RNA sequencing (RNA-Seq) to be associated with PTC metastatic profile. The pathway enrichment analysis was used to gain mechanistic insight into the gene panel generated from the genome-scale (omics) experiment. Considering the top 50 genes specifically deregulated in BRAF mutated PTC, we identified markers involved in multiple signaling pathways, including Wnt, focal adhesion, and cell-matrix interaction. Deregulation of members of these pathways have been identified to contribute to cancer metastasis in other tumor types. [36,37,38] and some are tightly regulated by the tumor surrounding microenvironment [39,40,41].

Our study reveals that PTC with mutated *BRAF*- (V600E) (but not *BRAF*-wild-type) presented a consistent overexpression of RSPO4, a member of the R-Spondin family of secretory proteins involved in the regulation of Wnt signaling. This family is comprised of RSPO1, RSPO2, RSPO3, and RSPO4; these members share overall 60% sequence homology and similar domain organization [34]. Despite the high structural homology between these members, the expression patterns and phenotypes in knockout mice have demonstrated striking differences [42]. In our cohort, from the four family members, only RSPO4 was significantly upregulated. This is the first report identifying *RSPO4* gene and protein overexpression in metastatic PTC. Earlier studies demonstrated that the stimulation of RSPO1–4 greatly potentiates the activity of Wnt ligands in both the canonical (B-catenin–dependent) and noncanonical (B-catenin–independent) pathways with major effects on the actin cytoskeleton, leading to loss of focal adhesions [43,44,45,46]. Furthermore, our study identified RSPO4 overexpression to be associated with BRAF V600E mutation, supporting a cooperation of these events to promote influence cell-matrix interaction, and cell motility and invasion. Interestingly, downregulation of *RSPO4* in cell lines induced a strong inhibition of activated forms of the focal adhesion-associated proteins including Fak, Paxillin, and Src which are major signaling molecules involved in the regulation cancer cell stiffness and cell locomotion/migration [47,48,49,50]. However, future studies are necessary to establish molecular mechanisms by which RSPO4 regulate these multiple intracellular targets and to exploit their potential therapeutic utility to develop inhibitors tailored for patients with advanced PTC that can also be exploited in combination with standard radiotherapy/chemotherapy to overcome resistance/recurrence.

## Figures and Tables

**Figure 1 cells-12-00139-f001:**
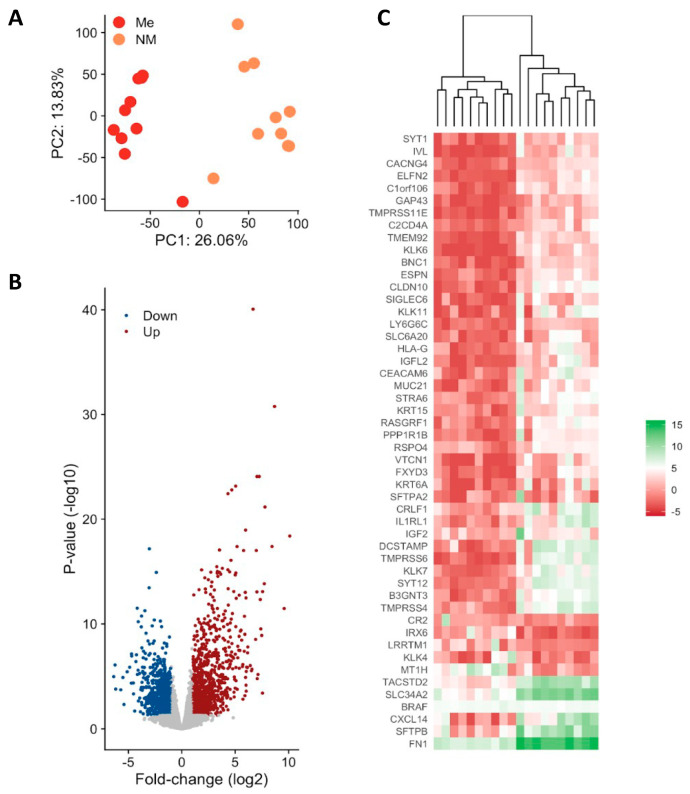
Differential gene expression in metastatic (Me) vs. non-metastatic (NM) PTC. (**A**) Principal component analysis (PCA). PCA plot of all RNA-Seq samples. Replicates of the same group are indicated by the same color as shown in the legend (**B**) Volcano plot of differentially expressed genes (**C**) Observed gene expression (log CPM) of all differentially expressed genes in all samples profiled shown as a Heatmap. Each column represents a sample, and each row represents a differentially expressed gene (DEGs). Upregulated genes are shown in red and downregulated genes are shown in blue. Scheme A was created using Biorender Free Software (https://biorender.com/, accessed on 1 January 2022).

**Figure 2 cells-12-00139-f002:**
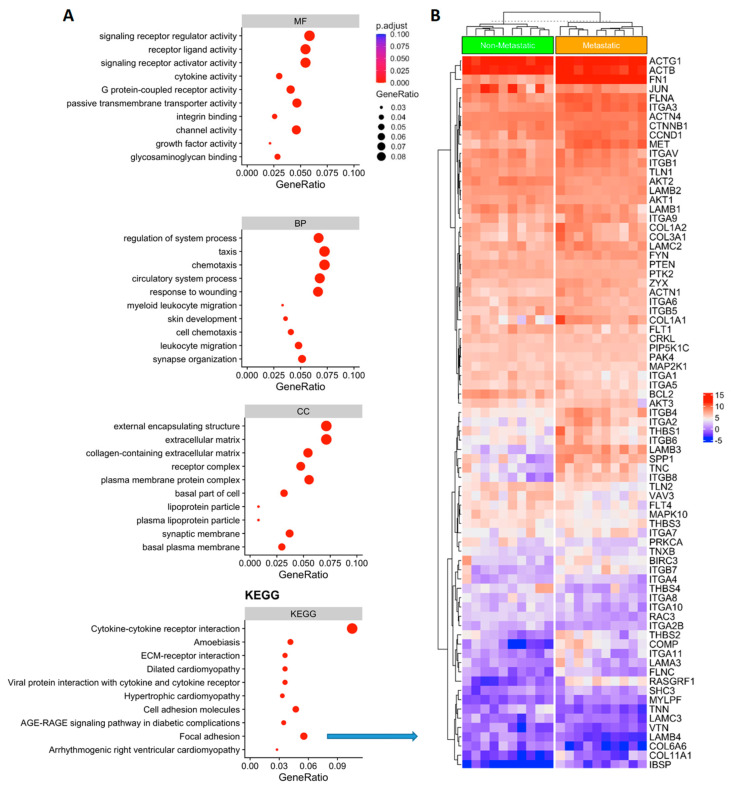
(**A**) Gene representation and gene set enrichment (GSEA) analysis of the differentially expressed genes. Gene Ontology (GO) enrichment analysis results showed the main differences regarding MF: molecular function; CC cellular component; and BP: biological process. KEGG pathways enrichment analysis (Kyoto Encyclopedia of Genes and Genomes) identified the focal adhesion gene network signaling with significant expression differences comparing metastatic vs. non metastatic thyroid cancer (KEEG; focal adhesion *p* ≤ 0.05). (**B**) Heatmap showing the focal adhesion gene expression in metastatic vs. non-metastatic PTC). Upregulated genes are shown in red and downregulated genes are shown in blue.

**Figure 3 cells-12-00139-f003:**
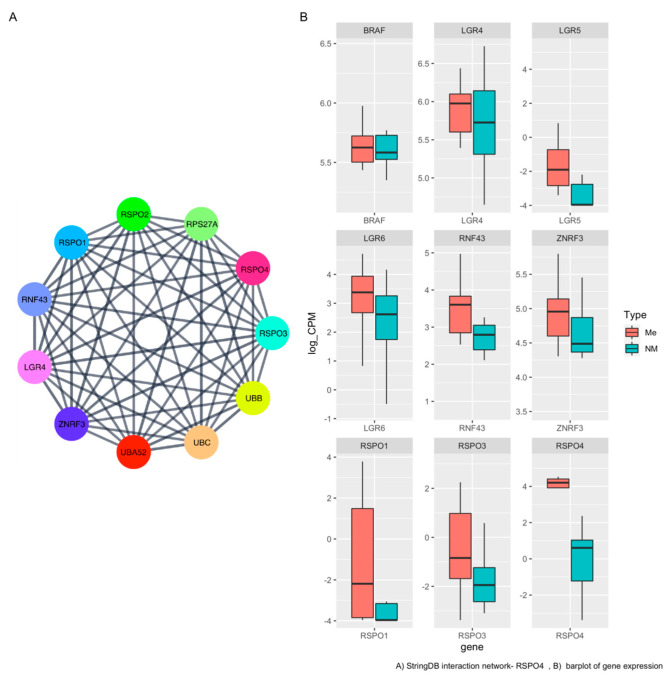
RSPO4 protein–protein interaction (PPI) network and expression level (**A**) Network graph of physical interacting partners of RSPO4 and the ligands using stringDB. (**B**) Representative boxplot of gene expression levels between metastatic and non-metastatic our PTC samples.

**Figure 4 cells-12-00139-f004:**
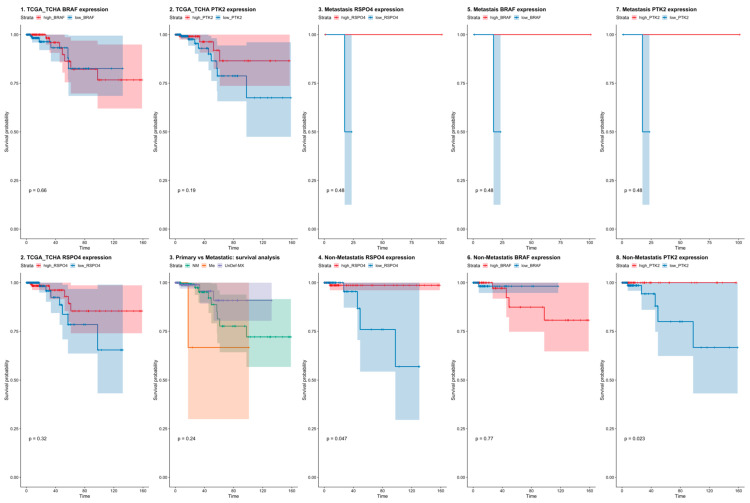
Survival rates in PTC patients from The Cancer Genome Atlas (TCGA) dataset considering BRAF, PTK2 (FAK), and RSPO4 gene expression. BRAF, PTK2 and RSPO4 expression levels discriminate between non-metastatic and metastatic PTC tumors with survival rates lower in PTC where these genes are overexpressed.

**Figure 5 cells-12-00139-f005:**
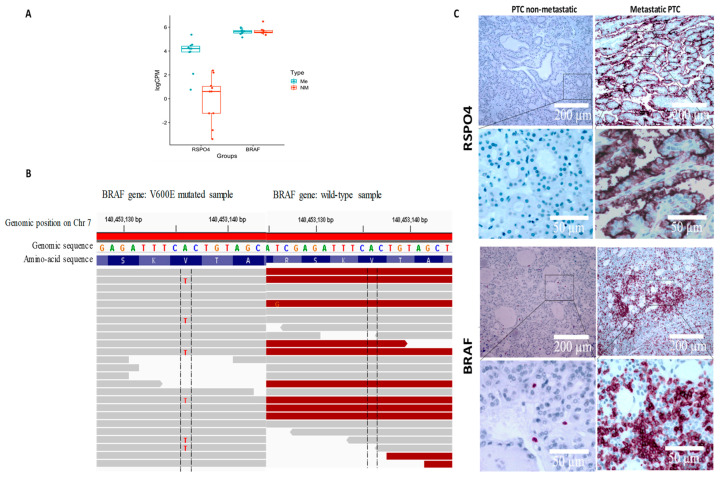
RSPO4 and BRAF genes and proteins were highly expressed in our metastatic cohort (**A**). Using Integrative Genome Viewer (IGV) all the 10 metastatic samples harboured an A to T substitution at position 140,453,136 on chromosome 7, that is harbouring a BRAF V600E classical mutation (**B**). Protein expression levels reveled that BRAF and RSPO4 are overexpressed in PTC samples with locoregional invasiveness (**C**).

**Figure 6 cells-12-00139-f006:**
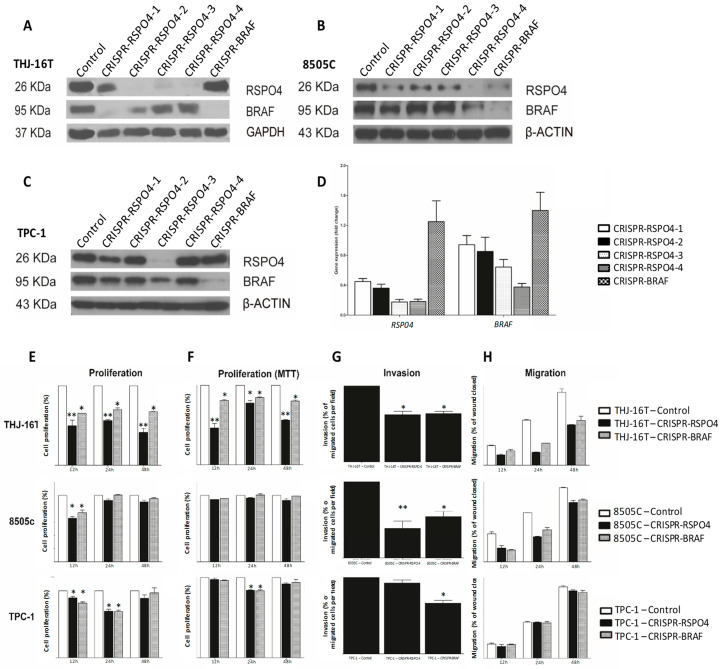
Identification of RSPO4 and BRAF sequence to gene knockout by CRISPR in metastatic and non-metastatic papillary thyroid cancer cell lines. THJ-16T (**A**), 8505c (**B**), and TPC-1 (**C**) cells were knockout using four different sequences for RSPO4 and one BRAF, which the protein and gene expression were verified by Western Blot (**A**–**C**) and qRT-PCR, respectively (**D**). (**A**) Western Blot of THJ-16T. (**B**) Western Blot of 8505c. (**C**) Western Blot of TPC-1. (**D**) RT-qPCR of THJ-16T relative to the control. The RSPO4 sequence 3 showed high efficiency to knockdown the gene all cell lines of the study. RSPO4 and BRAF knockdown decrease the proliferation, invasion, and migration in human thyroid cancer cell lines (**E**–**H**). After the knockout, CRISPR-RSPO4 metastatic cells (THJ-16T and 8505c) showed decrease in proliferation (measuring by cellular count and MTT assay), invasion, and migration. Error Bars represent SD from two three independent experiments. * *p* = 0.05, ** *p* < 0.01.

**Figure 7 cells-12-00139-f007:**
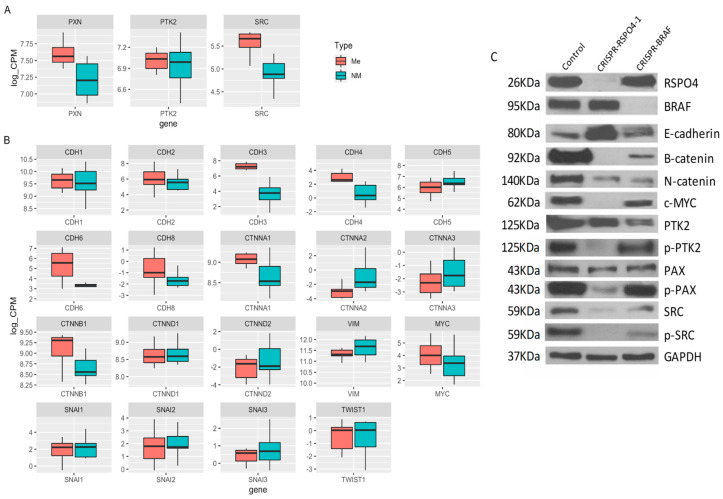
CRISPR-RSPO4 knockout reduce adhesion potential in PTC cell lines. (**A**) Box plot showing the gene expression data from PTC patient’s samples considering genes from focal adhesion network, (**B**) RSPO and BRAF signaling and (**C**) Cell lysates from control, CRISPR-RSPO4, and CRISPR-BRAF of metastatic cell line were immunoblotted with antibodies against Rspo4, Braf, E-cadherin, B-catenin, αN-catenin, Y69-phospho-cMyc, PTK2 (FAK), Y397-phospho-PTK2, Paxilin, 118Y-phospho-paxillin, Src, 416Y-phospho-Src, and Gapdh as an internal control. Error Bars represent SD from two experiments.

**Table 1 cells-12-00139-t001:** Clinicopathological characteristics of the patients with metastatic and non-metastatic PTC analyzed by RNAseq.

Group Variable	Non-Metastatic	Metastatic	*p*-Value *
Histological subtype	10/10 papillary	10/10 papillary	0.99
Age (mean, SE)	45.9 (4.7)	46.3 (4.8)	0.95
Gender (f/m)	7/3	6/4	0.99
pT category → (T3/T2-1)	4/6	4/6	0.99
pN category	pN0 (10/10)	pN+ (10/10)	<0.001

* Fisher’s exact for binary variable; *t*-test for continuous variable.

**Table 2 cells-12-00139-t002:** Sequences used for RSPO4 and BRAF knockdown.

Gene	NM	Clone
BRAF	NM_004333	TRCN0000231130
RSPO4-1	NM_001029871	TRCN0000139566
RSPO4-2	NM_001029871	TRCN0000139399
RSPO4-3	NM_001029871	TRCN0000139901
RSPO4-4	NM_001029871	TRCN0000139862

**Table 3 cells-12-00139-t003:** Specific primer set sequences to evaluate and validate the gene expression in this study.

Gene Symbol	Forward Sequence (5′–3′)	Reverse Sequence (5′–3′)
*ACTB*	GCACCCAGCACAATGAAG-	GCACCCAGCACAATGAAG-
*BRAF*	GAAGGTGAAGGTCGGA	GGGTCATTGATGGCAAC
*CDH1*	CCTGGGACTCCACCTACAGA	CCTGGGACTCCACCTACAGA
*CTNNB1*	ATTGTCCACGCTGGATTTTC	TCGAGGACGGTCGGACT
*GAPDH*	AATACACCAGCAAGCTAGATGC	AATCAGTTCCGTTCCCCAGAG
*HPRT1*	GAACGTCTTGCTCGAGATGTGA	TCCAGCAGGTCAGCAAAGAAT
*MYC*	ATCCAGCGTCTAAGCAGCTG	TACAACACCCGAGCAAGGAC
*PXN*	AGCTAGCGCGACCCTGA	TGTGGGAGGTGGTAGACTCC
*PTK2*	CCTGGTCCACTTGATCAGCTA	GCCAAAAGGATTTCTAAACCAG
*RSPO4*	GTGGAACAGCCGTTCTCCTCT	GAAGGAAGAAGCAAGTGGGC
*SNAIL*	CCAGTGCCTCGACCACTATG	CTGCTGGAAGGTAAACTCTGGA
*TWIST*	TCCATTTTCTCCTTCTCTGGAA	CCTTCTCGGTCTGGAGGAT
*VIM*	CTTCAGAGAGAGGAAGCCGA	ATTCCACTTTGCGTTCAAGG

**Table 4 cells-12-00139-t004:** Clinicopathological characteristics of the validation cohort from the PTC patients.

Characteristics	Benign Cases (*n* = 15)	PTC Matched from Seq Patients (*n* = 20)	PTC Long Follow-Up (*n* = 136)	All Patients (*n* = 171)	*p*-Value
**Age** (years, mean ± SD)	57.3 (14.6)	46.0 (14.7)	46.0 (14.4)	47.0 (14.6)	0.025
**Gender** Female Male	9 (60.0%) 6 (40.0%)	13 (65.0%) 7 (35.0%)	112 (82.4%) 24 (17.6%)	134 (78.4%) 37 (21.6%)	0.041
**Surgery** Total thyroidectomy Hemithyroidectomy	12 (80%) 3 (20%)	15 (75%) 5 (25%)	126 (92.6%) 10 (7.4%)	153 (89.5%) 18 (10.5%)	0.027
**pT category** pT1 pT2 pT3/4	na *	4 (20.0%) 5 (25.0%) 11(55.0%)	21 (15.4%) 33 (24.3%) 82 (60.3%)	25 (16.0%) 38 (24.4%) 93 (59.6%)	0.001
**pN category** pN0 pN1a pN1b pNx	na *	10 (50.0%) 6 (30.0%) 4 (20.0%) 0	74 (54.4%) 29 (21.3%) 18 (13.2%) 15 (11.0%)	84 (53.8%) 35 (22.4%) 22 (14.2%) 15 (9.6%)	0.001
**Follow-up** (months, mean ± SD)	14.5 (11.2)	15.7 (7.3%)	68.1 (30.9)	59.8 (34.5)	0.001

* na: not applicable.

## Data Availability

The datasets used and/or analyzed during the current study are available from the corresponding author on reasonable request.

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
