# Peer review of "Identification of R-Spondin Gene Signature Predictive of Metastatic Progression in BRAFV600E-Positive Papillary Thyroid Cancer"

_cells, 2022, doi:10.3390/cells12010139_

Round 1

Reviewer 1 Report

The work is interesting and undoubtedly relevant, but its disadvantage is that after conducting RNA-sequencing and obtaining a large amount of data, after many genes were presented that differ significantly in expression between Me and NM (Figure 1C), the authors chose only one gene, which does not look the most promising. As I understand it, the main advantage of this gene is that it has not been previously described in PTC.

In my opinion, it would be more logical to select several genes that differ most between Me and NM, validate the difference using PCR, and then check the best ones on cell lines. Otherwise, a large amount of data remains unused.

Remarks

Lines 52-53: Multi-omics experiments in thyroid cancer have allowed a comprehensive analysis and understanding of the molecular changes linked to tumor progression [12-16].

13. Alaoui-Jamali MA, Bismar TA, Gupta A, Szarek WA, Su J, Song W, Xu Y, Xu B, Liu G, Vlahakis JZ, Roman G, Jiao J, Schipper HM: A novel experimental heme oxygenase-1-targeted therapy for hormone-refractory prostate cancer. Cancer research 2009, 69(20):8017-8024. 579

15. Kowalski LP, Della Coletta R, Salo T, Maschietto M, Chojniak R, Lima JM, Mlynarek A, Hier MP, Alaoui-Jamali MA, Silva SD: Head and neck cancer: Emerging concepts in biomarker discovery and opportunities for clinical translation. Clinical and translational medicine 2020, 10(7). 583

The sentence refers to thyroid cancer, but the references refer to tumors of other localizations - they should be removed and replaced with more suitable ones.

Line 61: Clinicopathological predictive factors have shown only 4% success rate [9].

Where did 4% come from, in the cited publication there is no such thing.

Lines 65-71: All the metastatic PTC had BRAF mutation and up-regulation of R-spondin protein (RSPO4).

Preclinical studies using a panel of PTC and metastatic thyroid cancer cell lines supported that RSPO4 overexpression associated with BRAF V600E mutation can promote activation of focal adhesion kinases signaling in metastatic PTC.

The introduction should not report the results of the study.

Line 138: 10% 10% heat-inactivated foetal bovine serum (FBS)

10% Repeat

Lines 159-160: ACTB, GAPDH and HRPT1 was used as endogenous control.

It is necessary to clarify how normalization was carried out for three reference genes at once.

Line 186: 3x103 cells

10^(3) or 103

Line 203: Immunohistochemistry reaction was carried out on the TMA with 170 PTC samples [32].

32. de Lima JM, Castellano LRC, Bonan PRF, de Medeiros ES, Hier M, Bijian K, Alaoui-Jamali MA, da Cruz Perez DE, da Silva SD: Chitosan/PCL nanoparticles can improve anti-neoplastic activity of 5-fluorouracil in head and neck cancer through autophagy activation. The International Journal of Biochemistry & Cell Biology 2021, 134:105964.

Firstly, it is not clear why this reference is given here, this needs to be clarified.

Secondly, 170 PTC samples are mentioned here for the first time, although up to this point it seemed that there were only 20 samples, and then it turns out that not all of these 170 samples are PTC, and that there are 171 samples, not 170. The situation with samples should be clearly explained in paragraph “2.1. Study population”, perhaps add samples flow diagram.

Lines 227-230: For in vitro analysis, statistical analyses were performed using the two-tailed Student's t-test for unpaired samples to one independent experiments and One-way analysis of variance (ANOVA) with post-hoc comparisons based on the Tukey's multiple comparisons to two independent experiments.

Lines 234-237: For discrete variables showing normal distribution, means and standard errors of means (SEM) are given and comparisons were made using the t-test. Alternatively, median, first quartile (Q25), and third quartile (Q75) are indicated and the non-parametric Kruskal-Wallis test was used.

From everything written in this section, it remained unclear to me in which case, with the help of which test, statistical analysis was carried out, whether (by what method) the normality of the data distribution was determined, why in some cases the data representation in the form mean ± SD was used, and in some then as mean ± SEM.

Statistical analysis needs to be described more clearly.

Lines 245-246: The validation cohort constituted by a total of 171 patients with PTC plus 15 benign thyroid nodules and 20 PTC matched fresh samples

I wrote about this sample mess before, here it says 171 PTC patients + 15 benign thyroid nodules + 20 PTC matched fresh samples = 206 samples. And Table 4 indicates that there were 136 patients with PTC, 15 benign thyroid nodules + 20 PTC matched fresh samples = 171 samples.

The issue with samples needs to be clarified.

Line 252: The N-stage

Why is nothing written about M-stage? All samples M0?

Line 280: Supp Table 1

I would like to see this table; I did not find it among the files for download.

Figure 1.

The need for Figure 1A is questionable, I think that it should be removed.

Line 303: FA genes were clearly able to distinguish between metastatic vs. non-metastatic

Although everything is completely clear to the authors, it is still worth explaining in more detail why exactly FA genes were chosen, and not others, for example, CCRI?

Lines 319-320: In the unsupervisioned analysis (Figure 1D), a gene called RSPO4 was highly overexpressed in metastatic PTC.

In Figure 1D, this gene does not stand out among many other genes, it seems to me that a more detailed rationale for why this particular gene was chosen is needed.

Lines 360-362: …we used Integrative Genome Viewer (IGV) 37 to screened for common mutations described in thyroid cancer 13 to identify those impacted on our target genes identified to discriminate between non-metastatic and metastatic PTC cases.

What do the numbers 37 and 13 mean in this sentence?

Lines 369-370: BRAF V600E to BRAF wt papillary PTC found similar gene expression patterns (Figure 4).

It's probably Figure 5.

Figure 5. (B) Expression levels of BRAF and RSPO4. (C) BRAF and RSPO4 are overexpressed in PTC samples with locoregional invasiveness.

If I understand correctly, then the captions for the various parts of the figure are mixed up, while there is no caption for (A).

Figure 6

The quality of the picture is very poor, it is difficult to understand at least something.

Line 403: *P=0.05, **P<0.01, ***P<0.001.

P<0.05

Lines 446-447: It was identified 2034 differentially expressed genes (DEGs) by high-throughput RNA sequencing (RNA-Seq) involved in the metastatic PTC profile.

Suggestion: In these study 2034 differentially expressed genes (DEGs) was identified by high-throughput RNA sequencing (RNA-Seq) involved in the metastatic PTC profile.

Lines 447-451: Genes specifically deregulated in BRAF mutated PTC was further investigated to determine their implication for metastatic signaling and the top 50 DEGs were linked to focal adhesion, cytokine-cytokine interaction, and ECM interaction confirming the modulation of tumor microenvironment (TME) in the metastatic process [41,42].

It is not clear what the authors meant: that this study confirmed the modulation of TME in the metastatic process, or that the identified top 50 DEGs may be involved in this modulation, which has been described in other studies.

Paragraph starting on line 458: Dissecting the genetic landscape of the metastatic phenotype in PTC and…

This is a general discussion, not related to this study. Although there is nothing wrong with it, it should rather be in the Introduction to justify the purpose of the study.

Paragraph starting on line 469: Several newest therapies have been approved for the treatment of PTC.

This therapy part has little to do with the content of the study and would be best removed from the Discussion.

Author Response

REVIEWERS' COMMENTS

We thank the reviewers for their valuable comments. Below are point-by-point responses to each comments and the corrections made in the revised manuscript are tracked in yellow for the reviewers’ convenience).

REVIEWER #1:

General comment: The work is interesting and undoubtedly relevant, but its disadvantage is that after conducting RNA-sequencing and obtaining a large amount of data, after many genes were presented that differ significantly in expression between Me and NM (Figure 1C), the authors chose only one gene, which does not look the most promising. As I understand it, the main advantage of this gene is that it has not been previously described in PTC. In my opinion, it would be more logical to select several genes that differ most between Me and NM, validate the difference using PCR, and then check the best ones on cell lines. Otherwise, a large amount of data remains unused.

Response: We would like to stress that our primary focus on RSPO4 in this study was justified by: i) it is the most significantly regulated gene seen between metastatic and non-metastatic cases, (ii) its exclusive occurrence in BRAF V600 but not B-RAF wild type, and (iii) its impact on downstream focal adhesion signaling targets (e.g. FAK, Src, Paxillin…) known as drivers for cancer cell locomotion and invasion; a finding not described in current literature. Other genes we identified have marginal significance or been described in the literature as associated with progression in many cancers, including PTC. Regarding deep functional/molecular studies, we feel these lie beyond the scope of this study as these require time and extensive experiments given the increasing numbers of regulators involved in focal adhesion signaling as well as their crosstalks both with RAF and Wnt signaling.  Certainly, these studies are currently our priority using defined preclinical models. Last, we would like to mention that we have revised the discussion accordingly and revised the title to best describe the objective of our work.

Specific comments

Comment 1:

Lines 52-53: Multi-omics experiments in thyroid cancer have allowed a comprehensive analysis and understanding of the molecular changes linked to tumor progression [12-16].

  1. Alaoui-Jamali MA, Bismar TA, Gupta A, Szarek WA, Su J, Song W, Xu Y, Xu B, Liu G, Vlahakis JZ, Roman G, Jiao J, Schipper HM: A novel experimental heme oxygenase-1-targeted therapy for hormone-refractory prostate cancer. Cancer research 2009, 69(20):8017-8024. 579
  2. Kowalski LP, Della Coletta R, Salo T, Maschietto M, Chojniak R, Lima JM, Mlynarek A, Hier MP, Alaoui-Jamali MA, Silva SD: Head and neck cancer: Emerging concepts in biomarker discovery and opportunities for clinical translation. Clinical and translational medicine 2020, 10(7). 583

The sentence refers to thyroid cancer, but the references refer to tumors of other localizations - they should be removed and replaced with more suitable ones.

Response: These references were replaced accordingly (page 17; lines 548-553).

  1. Rossi ED, Locantore P, Bruno C, Dell'Aquila M, Tralongo P, Curatolo M, Revelli L, Raffaelli M, Larocca LM, Pantanowitz L, Pontecorvi A. Molecular Characterization of Thyroid Follicular Lesions in the Era of "Next-Generation" Techniques. Front Endocrinol (Lausanne). 2022, (13):834456.

  1. Boufraqech M, Nilubol N. Multi-omics Signatures and Translational Potential to Improve Thyroid Cancer Patient Outcome. Cancers (Basel). 11 (12): 1988.

Comment 2:

Line 61: Clinicopathological predictive factors have shown only 4% success rate [9].

Where did 4% come from, in the cited publication there is no such thing.

Response:  We apologize for this confusion due to inadvertent textual truncation, which should read 4% to 28% [9]. We have made the correction (page 2, line 61), which now read: Studies analyzing PTC of all sizes described recurrence rates ranging from 4% to 28% [9].

Comment 3:

Lines 65-71: All the metastatic PTC had BRAF mutation and up-regulation of R-spondin protein (RSPO4).

Preclinical studies using a panel of PTC and metastatic thyroid cancer cell lines supported that RSPO4 overexpression associated with BRAF V600E mutation can promote activation of focal adhesion kinases signaling in metastatic PTC.

The introduction should not report the results of the study.

Response: We agree. We have revised the introduction section accordingly.

Comment 4:

Line 138: 10% 10% heat-inactivated foetal bovine serum (FBS)

10% Repeat

Response: This duplication was removed (page 4, line 137).

Comment 5:

Lines 159-160: ACTB, GAPDH and HRPT1 was used as endogenous control.

It is necessary to clarify how normalization was carried out for three reference genes at once.

 Response: The gene normalization used geNorm, a widely used algorithm to determine the most stable internal control (housekeeping) genes from a set of tested candidate reference genes in each sample panel. Gene expression normalization factor is then calculated for each sample based on the geometric mean of a user-defined number of reference genes (https://genorm.cmgg.be/). We added a new sentence in the sub-topic of the methods section #2.7 to clarify this aspect. qRT-PCR is used to determine gene normalization among samples (Page 4, lines 159-162). In the revised manuscript the following description was added: GeNorm (https://genorm.cmgg.be/) algorithm was used to determine the most stable internal reference genes. The software packages were used as excel add-ons. ACTB, GAPDH and HRPT1 were selected by the algorithm as internal controls.

Comment 6:

Line 186: 3x103 cells

10^(3) or 103

 Response: Correction was made: 3x103 cells (Page 5, line 187).

Comment 7:

Line 203: Immunohistochemistry reaction was carried out on the TMA with 170 PTC samples [32].

  1. de Lima JM, Castellano LRC, Bonan PRF, de Medeiros ES, Hier M, Bijian K, Alaoui-Jamali MA, da Cruz Perez DE, da Silva SD: Chitosan/PCL nanoparticles can improve anti-neoplastic activity of 5-fluorouracil in head and neck cancer through autophagy activation. The International Journal of Biochemistry & Cell Biology 2021, 134:105964.

Firstly, it is not clear why this reference is given here, this needs to be clarified.

Secondly, 170 PTC samples are mentioned here for the first time, although up to this point it seemed that there were only 20 samples, and then it turns out that not all of these 170 samples are PTC, and that there are 171 samples, not 170. The situation with samples should be clearly explained in paragraph “2.1. Study population”, perhaps add samples flow diagram.

 Response: We apologize for this confusion. The citation was modified (page 18).

For PTC samples, the total number of cases evaluated in our study was 171 cases (Table 4). Correction was made (page 5, lines 204).

Comment 8:

Lines 227-230: For in vitro analysis, statistical analyses were performed using the two-tailed Student's t-test for unpaired samples to one independent experiments and One-way analysis of variance (ANOVA) with post-hoc comparisons based on the Tukey's multiple comparisons to two independent experiments.

Lines 234-237: For discrete variables showing normal distribution, means and standard errors of means (SEM) are given and comparisons were made using the t-test. Alternatively, median, first quartile (Q25), and third quartile (Q75) are indicated and the non-parametric Kruskal-Wallis test was used.

From everything written in this section, it remained unclear to me in which case, with the help of which test, statistical analysis was carried out, whether (by what method) the normality of the data distribution was determined, why in some cases the data representation in the form mean ± SD was used, and in some then as mean ± SEM.

Statistical analysis needs to be described more clearly.

  Response:  The standard deviation (SD) measures the amount of variability, or dispersion, from the individual data values to the mean. Standard error of the mean (SEM) measures how far the sample mean (average) of the data is likely to be from the true population mean. We have revised the statistical analysis section to provide more clarity; changes are highlighted (page 6, lines 231-234; 236; 238-240).

Comment 9:

Lines 245-246: The validation cohort constituted by a total of 171 patients with PTC plus 15 benign thyroid nodules and 20 PTC matched fresh samples

I wrote about this sample mess before, here it says 171 PTC patients + 15 benign thyroid nodules + 20 PTC matched fresh samples = 206 samples. And Table 4 indicates that there were 136 patients with PTC, 15 benign thyroid nodules + 20 PTC matched fresh samples = 171 samples.

The issue with samples needs to be clarified.

 Response: We apologize for the confusion. As shown in the Table 4, the total number of cases evaluated in our study was 171 cases (5th column called all patients n=171). Table 4 is the clinicopathological characterization of the benign cases (n=15); PTC specimens corresponding to the frozen tissues used for RNAseq experiments for experimental validation (n=20); and PTC with long term follow-up (n=136) for biological validation (page 7). We added an additional clarification on page 7, lines 249-251 as follow:

The validation cohort constituted by a total of 171 patients with PTC, being 15 benign thyroid nodules and 20 PTC matched fresh samples (10 metastatic and 10 non-metastatic) and 136 PTCs with long-term follow-up.

Comment 10:

Line 252: The N-stage

Why is nothing written about M-stage? All samples M0?

  Response:  All the patients had no distant metastasis at initial diagnosis.

Comment 11:

Line 280: Supp Table 1

I would like to see this table; I did not find it among the files for download.

  Response:  Supplemental table is provided in the supplemental material section.

Comment 12:

Figure 1.

The need for Figure 1A is questionable, I think that it should be removed.

  Response: Figure 1A was removed (page 8).

Comment 13:

Line 303: FA genes were clearly able to distinguish between metastatic vs. non-metastatic

Although everything is completely clear to the authors, it is still worth explaining in more detail why exactly FA genes were chosen, and not others, for example, CCRI?

   Response: To identify the major signaling pathways involved in metastatic tumors, we further analyzed DEGs for KEGG pathway enrichment. In our analysis, focal adhesion (FA) signaling network was enriched in addition to cytokine–cytokine interaction pathway (CCRI). However, we observed that FA genes overlapped among more than one network, e.g., cell-cell adhesion and ECM-receptor interaction pathways (Figure 2A). We added this information in the revised version of our manuscript (page 9, lines 291-300), as follow:

The GO and KEEG gene set enrichment pathways analysis revealed the largest gene ratio differences in molecular function (MF), cellular component (CC), biological process (BP) and signaling pathways are linked to cytokine-cytokine receptor interaction (CCRI) and focal adhesion signaling (FA) (Figure 2A-B).  When we combined MF, CC, BP and KEEG analysis, FA genes were overlapped in more than one network, such as the cell adhesion molecules and ECM-receptor interaction pathways (Figure 2A). These genes were clearly able to distinguish between metastatic vs. non-metastatic groups (Figure 2B), suggesting that FA process had disrupted key genes in metastatic PTC. Hence, we decided to focus on this pathway and analyze the metastatic potential in PTC.

Comment 14:

Lines 319-320: In the unsupervised (Figure 1D), a gene called RSPO4 was highly overexpressed in metastatic PTC.

In Figure 1D, this gene does not stand out among many other genes, it seems to me that a more detailed rationale for why this particular gene was chosen is needed.

   Response: Figure 1D shows a list of the top 50 DEGs and, in general, any of these genes could be investigated because they are differently expressed between metastatic and non-metastatic (<5% FDR, P<0.05, log2FC>|2|). However, this is the reason that in gene expression analysis, we can not limit our selection based only on the heatmap. In our study, DEGs were constantly tested and submitted to Gene ontology (GO) and Kyoto Encyclopedia of Genes and Genomes (KEGG) enrichment analyses to target specifically a gene with essential role in pathways and networks that are overlapping. We highlighted this information in the results (page 10, lines 313-316) as follow:

In the unsupervisioned analysis (Figure 1D), a gene called RSPO4 was highly overexpressed in metastatic PTC (<5% FDR, P<0.05, log2FC>|2|). Based on the lists of pathways and networks potentially enriched in metastatic PTC (Figure 2A). RSPO4 showed to be a promising candidate

Comment 15:

Lines 360-362: …we used Integrative Genome Viewer (IGV) 37 to screened for common mutations described in thyroid cancer 13 to identify those impacted on our target genes identified to discriminate between non-metastatic and metastatic PTC cases.

What do the numbers 37 and 13 mean in this sentence?

  Response: We appreciate your careful revision. These numbers were supposed to be the original reference to the IGV tool. However, we realized that the reference is using an outdated version of the integrative tool. So, the best way to cite this reference tool is by the website where the update versions are available and a list of publications is cited there too. So, we added this information in the revised version of our manuscript (page 12, lines 354-358) as follow:

Since both RSPO4 and BRAF gene was highly expressed in our metastatic cohort (Figure 5A), we used Integrative Genome Viewer (IGV - https://software.broadinstitute.org/software/igv/) to screened for common mutations described in thyroid cancer to identify those impacted on our target genes identified to discriminate between non-metastatic and metastatic PTC cases.

Comment 16:

Lines 369-370: BRAF V600E to BRAF wt papillary PTC found similar gene expression patterns (Figure 4).

It's probably  Figure 5.

   Response: The reference to the Figure 5 was now added in the revised version (page 12; line 365).

Comment 17:

Figure 5. (B) Expression levels of BRAF and RSPO4. (C) BRAF and RSPO4 are overexpressed in PTC samples with locoregional invasiveness.

If I understand correctly, then the captions for the various parts of the figure are mixed up, while there is no caption for (A).

   Response: The legend was re-written to avoid any misunderstand in the revised version of our manuscript (Page 13, lines 368-372).

Comment 18:

Figure 6

The quality of the picture is very poor, it is difficult to understand at least something.

Response: We apologize because the quality of the Figure 6 was lost when we have to reduce the size of the original file to upload the paper in the system. Now, this issue was corrected and a better quality of the picture is available in the revised version of our manuscript (page 14).

Comment 19:

Line 403: *P=0.05, **P<0.01, ***P<0.001.

P<0.05

    Response:  When presenting P values some groups find it helpful to use the asterisk rating system as well as quoting the P value:

P = 0.05 *, P < 0.01 **, P < 0.001***

Most authors refer to statistically significant as P = 0.05 and statistically highly significant as P < 0.001 (less than one in a thousand chance of being wrong). The asterisk system avoids the woolly term "significant".

Comment 20:

Lines 446-447: It was identified 2034 differentially expressed genes (DEGs) by high-throughput RNA sequencing (RNA-Seq) involved in the metastatic PTC profile.

Suggestion: In this study 2034 differentially expressed genes (DEGs) was identified by high-throughput RNA sequencing (RNA-Seq) involved in the metastatic PTC profile.

     Response:  The text was modified as suggested (page 16, lines 439-440) as follow:

2034 differentially expressed genes (DEGs) were identified by high-throughput RNA sequencing (RNA-Seq) to be associated with PTC metastatic profile.

Comment 21:

Lines 447-451: Genes specifically deregulated in BRAF mutated PTC was further investigated to determine their implication for metastatic signaling and the top 50 DEGs were linked to focal adhesion, cytokine-cytokine interaction, and ECM interaction confirming the modulation of tumor microenvironment (TME) in the metastatic process [41,42].

It is not clear what the authors meant: that this study confirmed the modulation of TME in the metastatic process, or that the identified top 50 DEGs may be involved in this modulation, which has been described in other studies.

  Response: The pathway enrichment analysis helps us to gain mechanistic insight into the gene list generated from the genome-scale (omics) experiment. Considering the top 50 genes specifically deregulated in BRAF mutated PTC, we were able to identify the biological pathways (e.g., focal adhesion, cytokine-cytokine interaction, and ECM interaction) involved with tumor microenvironment (TME) modulation. These pathways have also been showed to be deregulated during the metastatic process in different tumors.

In order to clarify this information, the sentence was re-written (page 16, lines 440-447) as follow:

The pathway enrichment analysis was used to gain mechanistic insight into the gene panel generated from the genome-scale (omics) experiment. Considering the top 50 genes specifically deregulated in BRAF mutated PTC, we identified markers involved in multiple signaling pathways, including Wnt, focal adhesion, and cell-matrix interaction. Deregulation of members of these pathways have been identified to contribute to cancer metastasis in other tumor types. [41-43] and some are tightly regulated by the tumor surrounding microenvironment [45,46].

Comment 22:

Paragraph starting on line 458: Dissecting the genetic landscape of the metastatic phenotype in PTC and…

This is a general discussion, not related to this study. Although there is nothing wrong with it, it should rather be in the Introduction to justify the purpose of the study.

Response: Modification was made (page 16).

Comment 23:

Paragraph starting on line 469: Several newest therapies have been approved for the treatment of PTC.

This therapy part has little to do with the content of the study and would be best removed from the Discussion.

Response: We removed this paragraph as you suggested (page 16).

Reviewer 2 Report

This study investigated papillary thyroid carcinoma (PTC) cases with an unusual aggressive phenotype using a high-throughput RNA sequencing to identify differentially regulated genes associated with metastatic PTC. And authors concluded RSPO4 overexpression associated with BRAF V600E mutation promotes activation of multiple members of focal adhesion signaling network in metastatic PTC.

RSPO proteins constitute a family of four secreted glycoproteins (RSPO1-4) that have appeared as multipotent signaling ligands. The best-known molecular function of RSPOs lies within their capacity to agonize the Wnt/β-catenin signaling pathway. However, so far, the role and related signaling pathway of RSPO in thyroid cancer have not been reported. Therefore, it is very important and interesting topic.

I ask that authors should revise it based on the below commentary.

(1) The purpose of this study is to investigate the role of RSPO in PTC. Delete the benign cases (n=15) in Table 4. And, recently, the proportion of papillary thyroid microcarcinoma (PTMC) in thyroid cancer patients is increasing. Why did you exclude PTMC cases in this study?

(2) In this study, does metastatic PTC mean locoregional or distant metastasis?

(3) In line 249 and 250, correct the ‘times’ to ‘cases’. And match the contents of line 252-254 with the table 4.

(4) The contents of table 1 and 4 do not match. T2-1 cases are 12 in table 1 but 9 in table 4 (PTC matched from Seq patients). And, according to table 4 (PTC matched from Seq patients), did you include Nx in N0? Nx and N0 are completely different concepts, so please analyze again with N0 cases.

(5) In general, it is known that BRAF mutation in PTC is related to aggressiveness. In Figure 3(B), there is no difference in BRAF expression between metastatic and non-metastatic PTC. Why? Can you explain it?

(6) The content of Figure 6 cannot be confirmed because the resolution is not good. Please present Figure 6 again with a good resolution.

(7) According to Figure 7 (A), there is no difference in PTK2 expression between metastatic and non-metastatic PTC. Please add an explanation of this result.

(8) In Figure 7 (C), N-catenin is αN-catenin or N-cadherin? The Figure and the legend do not match. And there is no result for the vimentin in the legend. Please check it.

(9) In ‘Discussion’, please add more details about the results of Figure 4 and 7.

(10) In ‘Conclusions’, the 1st paragraph (line 515-518) is not related to this study. Please delete it.

(11) Please revise Keywords to MESH term.

Author Response

REVIEWER #2:

This study investigated papillary thyroid carcinoma (PTC) cases with an unusual aggressive phenotype using a high-throughput RNA sequencing to identify differentially regulated genes associated with metastatic PTC. And authors concluded RSPO4 overexpression associated with BRAF V600E mutation promotes activation of multiple members of focal adhesion signaling network in metastatic PTC.

RSPO proteins constitute a family of four secreted glycoproteins (RSPO1-4) that have appeared as multipotent signaling ligands. The best-known molecular function of RSPOs lies within their capacity to agonize the Wnt/β-catenin signaling pathway. However, so far, the role and related signaling pathway of RSPO in thyroid cancer have not been reported. Therefore, it is very important and interesting topic.

I ask that authors should revise it based on the below commentary.

Comment 1:

(1) The purpose of this study is to investigate the role of RSPO in PTC. Delete the benign cases (n=15) in Table 4. And, recently, the proportion of papillary thyroid microcarcinoma (PTMC) in thyroid cancer patients is increasing. Why did you exclude PTMC cases in this study?

Response: Thyroid papillary microcarcinoma (PTMC) is a subtype of papillary carcinoma that included tumors with less than 10mm diameter. These small tumors are surround by fibroblasts, vessels and different inflammatory cell population. In our study, we decided to guarantee a homogenous cell population to be submitted to RNAseq in order to retrieve the differentially expressed genes (DEGs) specifically from the tumor cells (not contaminated by the diverse type of cells from the microenvironment), and consequently not compromise the data related to tumor cells. It will be fantastic to design a new study focusing on PTMC where the samples could be submitted to laser capture microdissection (LCM) and have the DEGs information only from a homogeneous tumor cells populations (not cell from the TME). The begin cases were used as control for IHC reaction.

Comment 2:

(2) In this study, does metastatic PTC mean locoregional or distant metastasis?

Response: The primary goal of our study was to identify transcriptomic differences in a unique cohort of rare PTCs with an unusual aggressive phenotype presenting with extensive lymph node metastasis and higher incidence of locoregional recurrence (metastatic), compared to a non-metastatic PTC. Metastatic PTC means that patients had locoregional metastasis. We added the metastatic word in parenthesis to clarify it in the introduction (page 2, line 65).

Comment 3:

(3) In line 249 and 250, correct the ‘times’ to ‘cases’. And match the contents of line 252-254 with the table 4.

Response: The word was corrected as you suggested (page 7, line 255) and the contexts were matched with Table 4 (page 7, lines 253-259) as follow:

Total thyroidectomy was performed in 153 patients (89.5%), meanwhile hemithyroidectomy was performed in 18 cases (10.5%). Ninety-three (59.6%) of the patients had pT3-4 tumors, meanwhile 63 had pT1-2 (40.4%) tumors. Average tumor size was 2.6cm (SD ±1.53). Any extrathyroidal extension was noted in 76 (44.4%) cases. The N-stage was pN0 in 84 (53.8%) cases, pN1a in 35 (22.4%) cases, pN1b in 22 (14.2%) cases, and pNx in 15 (9.6%) cases, and not available in 15 cases (benign thyroid tissue cases) (Table 4).

Comment 4:

(4) The contents of table 1 and 4 do not match. T2-1 cases are 12 in table 1 but 9 in table 4 (PTC matched from Seq patients). And, according to table 4 (PTC matched from Seq patients), did you include Nx in N0? Nx and N0 are completely different concepts, so please analyze again with N0 cases.

Response: Table 1 is the distribution of the clinicopathological characteristics of the patients with metastatic and non-metastatic PTC submitted to RNAseq to show how the samples were matched by clinical stage (T and N), histological subtype, age and gender in order to avoid bias during the analysis of metastatic versus non-metastatic. This was now clarified directly in the title of the table in the page 3 (lined 101-102). Please note that the cases were matched by T3-2-1 (Table 1) and Table 4 is representing (T1+T2 versus T3+T4), so, the number of cases distributed in these groups are slightly different. In relation to the lymph nodes status, the analysis was done comparing pN0 versus pN+ (pN1a + pN1b). This data was also highlighted in the Table 4 that is representing the general distribution of all cases used for RNAseq and for large validation (TMA) (page 7).

Comment 5:

(5) In general, it is known that BRAF mutation in PTC is related to aggressiveness. In Figure 3(B), there is no difference in BRAF expression between metastatic and non-metastatic PTC. Why? Can you explain it?

Response: Mutations in BRAF, most frequently the valine (V) to glutamate (E) substitution at residue 600 (V600E) are considered cancer-initiating mutations in PTC. However, identical mutations are not always expressed at RNA levels in all individuals who carry it; moreover, when a mutation is expressed, it is not always expressed at the level same level between similar cases. Several factors can contribute to these discrepancies. These include (i) the notorious inter- and intra-cellular heterogeneity seen in most solid tumours, including PTC; (ii) factors related to gene penetrance (proportion of a population of individuals who carry a disease-causing allele and express the related disease phenotype ) and expressivity (phenotypic expression). We would like to also highlight that aberrantly spliced isoforms of BRAF V600E (BRAF V600E DEx) have been identified in malignant cells. Certainly, single cell sequencing, which we are planning in our future studies, can help address some of these discrepancies.

Comment 6:

(6) The content of Figure 6 cannot be confirmed because the resolution is not good. Please present Figure 6 again with a good resolution.

Response: We apologize because the quality of the Figure 6 was lost when we have to reduce the size of the original file to upload the paper in the system. Now, this issue was corrected and a better quality of the picture is available in the revised version of our manuscript (page 14).

Comment 7:

(7) According to Figure 7 (A), there is no difference in PTK2 expression between metastatic and non-metastatic PTC. Please add an explanation of this result.

     Response:  Considering our samples, PTK2 was overexpressed in thyroid cancer (please see control group Figure 7C). This protein-tyrosine kinase plays an essential role in regulating cell migration, adhesion, spreading, reorganization of the actin cytoskeleton, formation and disassembly of focal adhesions and cell protrusions, cell cycle progression, cell proliferation and apoptosis. PTK2 protein expression decreased when the thyroid cancer cells were treated with CRISPR-RSPO4 and CRISPR-BRAF. We also observed that focal adhesion kinase is phosphorylated and activated in thyroid cancer (control) and CRISPR-BRAF, but not in CRISPR-RSPO4 showing that the autophosphorylation of FAK Y397 is not required for FAK activation in the control and CRISPR-BRAF. We modified the text in the revised version of our manuscript (pages 14 and 15) as follow:

3.8. RSPO4 inhibition in PTC cells prevents the activation of proteins involved in focal adhesion signaling

Metastatic and non-metastatic thyroid cancer cell lines carrying CRISPR-RSPO4 knockout, and their controls were used to understand how the gene modulation could influence the phenotypic changes related to cell movement and invasion. As expected, inhibition of RSPO4 induced a downregulation of Wnt/catenins and an upregulation of E-cadherins. Interestingly, RSPO4 downregulation resulted in a potent inhibition of activated forms of mutiple phosphorylated proteins involved in focal adhesion signaling (Figure 7A-C). These include Y397 (p-PTK2), the autophosphorylation site of focal adhesion kinase (PTK2/FAK) critical for PTK2 activation and interaction with downstream partners such as 118Y-p-Paxillin, an adapter proteins and partner of PTK2 in focal adhesion signaling and regulation of focal adhesion turnover during cancer cell locomotion; and 416Y-p-Src, which regulate Src multiple downstream signaling including its interaction with PTK2 during in cell migration and invasion.

Comment 8:

(8) In Figure 7 (C), N-catenin is αN-catenin or N-cadherin? The Figure and the legend do not match. And there is no result for the vimentin in the legend. Please check it.

     Response: Figure 7 C is referring to αN-catenin, synthetized by CTNNA1 gene. The legend was reviewed to guarantee that there is no confusion (Page 15, lines 419-427) as follow:

Figure 7: CRISPR-RSPO4 knockout reduce adhesion potential in PTC cell lines. (A) Box plot showing the gene expression data from PTC patient’s samples considering genes from focal adhesion network, (B) RSPO and BRAF signaling and (C) Cell lysates from control, CRISPR-RSPO4, and CRISPR-BRAF of metastatic cell line were immunoblotted with antibodies against Rspo4, Braf, E-cadherin, B-catenin, αN-catenin, Y69-c-Myc, PTK2 (FAK), p-PTK2, Paxilin, 118Y-p-paxillin, Src, 416Y-p-Src, and Gapdh as an internal control. Error Bars represent SD from two experiments. *P<0.05, **P<0.01, ***P<0.001. Analyses performed against control cells (*) or cells with knockout genes (RSPO4 and BRAF).

Comment 9:

(9) In ‘Discussion’, please add more details about the results of Figure 4 and 7.

     Response:  We added a new paraph explaining in details the Figure 7 (pages 14 and 15) as follow:

Metastatic and non-metastatic thyroid cancer cell lines carrying CRISPR-RSPO4 knockout, and their controls were used to understand how the gene modulation could influence the phenotypic changes related to cell movement and invasion. As expected, inhibition of RSPO4 induced a downregulation of Wnt/catenins and an upregulation of E-cadherins. Interestingly, RSPO4 downregulation resulted in a potent inhibition of activated forms of mutiple phosphorylated proteins involved in focal adhesion signaling (Figure 7A-C). These include Y397 (p-PTK2), the autophosphorylation site of focal adhesion kinase (PTK2/FAK) critical for PTK2 activation and interaction with downstream partners such as 118Y-p-Paxillin, an adapter proteins and partner of PTK2 in focal adhesion signaling and regulation of focal adhesion turnover during cancer cell locomotion; and 416Y-p-Src, which regulate Src multiple downstream signaling including its interaction with PTK2 during in cell migration and invasion.

Comment 10:

(10) In ‘Conclusions’, the 1st paragraph (line 515-518) is not related to this study. Please delete it.

     Response:  The text was revised and major modifications were done in the discussion/conclusion topic (pages 15 and 16).

Comment 11:

(11) Please revise Keywords to MESH term.

     Response:  We modified the keywords using now the Medical Subject Headings (MeSH) terms. So, instead of Papillary thyroid cancer; RSPO4; BRAF V600E; focal adhesions; prognosis, now it is written: Thyroid Cancer, Papillary, R-spondin 4, RSPO, BRAF, Prognoses (page 1).

Round 2

Reviewer 2 Report

It’s well corrected by reviewer’s opinion.

Thank you for your efforts.